# How does fintech affect bank efficiency in Taiwan?

**Chang-Sheng Liao** [ID] *

School of Economics and Management, Hubei Polytechnic University, Hubei, China

* sheng2009tw@gmail.com

**Data Availability Statement:** All relevant data are within the paper and its Supporting information files.

**Funding:** This study funders support by below Fujian Association for Science and Technology, (No.2020R0166) awarded to CSL and Hubei

## Abstract

This study investigates the impact of financial technology (fintech) on bank efficiency in Taiwan from 2007 to 2020, using parametric and non-parametric analyses. In the future, fintech is expected to play an important role in bank sustainability operations. The findings show that fintech-related initiatives affect bank efficiency. These results illustrate how fintech can help banks directly or indirectly improve their efficiency, leading to higher profits or lower costs. For instance, mobile payment systems, a fintech-related initiative, improve bank efficiency. However, this degree of improvement is not substantial. In general, the findings imply that fintech-related initiatives do not significantly improve bank efficiency, implying an IT "productivity paradox."

## Introduction

Information technology (IT) boosts firms' efficiency, reduces resource input, and increases output. Fintech is a portmanteau of "financial technology." It refers to new technology that aims to improve and automate the delivery and use of financial services. That is, it is an emerging industry that significantly uses IT to improve the financial and banking sectors. It enables banks to exploit innovative marketing and sales channels to promote product awareness to existing and potential customers. It also helps enhance business models, financial products, and processes. It creates a significant effect on financial markets and institutions and the provision of financial services. Numerous studies have shown that IT can significantly impact supply chain management, enterprise resource planning systems, product innovation performance, and organizational collaboration. However, few studies have examined whether the impact of fintech on bank efficiency is significantly positive or negative.

Arner et al. [1] and Chen et al. [2] argued that fintech has three phases. Fintech 1.0, which existed before 1967, was an analog industry. Fintech 2.0, which lasted from 1967 to 2008, could be characterized by new delivery channels, automated teller machines (ATMs), online banking, telephone banking, and non-cash payment (e.g., credit card). Notably, internet financial service enterprises have arisen since 2009. Traditional banks have started to use more fintech-related initiatives to provide digital products, such as mobile payment systems. Furthermore, new fintech-related initiatives, such as big data and blockchain, have improved banks' fund efficiency and internet trading security.

Understanding the impact of IT (Fintech) on bank efficiency is important because it incurs high costs and influences bank operations and strategies (Beccalli, [3]). In doing so, this study

Polytechnic University, No.23xjr12R awarded to CSL.

**Competing interests:** The authors have declared that no competing interests exist.

investigates whether fintech influences bank efficiency. The frontier literature incorporates variables related to IT in the evaluation of bank efficiency. Evaluating bank efficiency by incorporating IT-related variables are considered true bank efficiency measures (Cook et al., [4]). In addition, Cho and Chen [5] show that the greater the banks' presence of mobile devices and third-party payment transaction volume, the higher the cost productivity growth rate. Lee et al. [6] find that Fintech development and innovation improve bank efficiency and the technology banks use. Furthermore, Lee et al. [7] posit that Fintech development reduces commercial banks' overall efficiency. However, these studies do not consider Fintech evolution or the different stages in Fintech that affect bank efficiency. Thus, here we provide an interesting case study on the effects of Fintech initiatives, such as ATMs (Fintech 2.0), mobile payment (Fintech 3.0), and IT investment.

The contributions of this study are as follows. First, few studies have considered Fintech-related initiatives in the context of bank efficiency, with some evaluating Fintech initiatives using linear regression. This is the first study to consider Fintech-related initiatives in connection to bank efficiency in Taiwan, which assumes Fintech as product output. Second, to the best of my knowledge, this study is the first to explain the difference impacts of Fintech 2.0 and Fintech 3.0 on bank efficiency. Financial products and services of Fintech 2.0 include ATMs and telephone banking, and those of Fintech 3.0 include mobile payment and blockchain services. Third, previous literature has often used a single method to evaluate bank efficiency; therefore, this study employs parametric and non-parametric approaches to evaluate bank efficiency and provides a comprehensive analysis of the impact of Fintech on bank efficiency. Fourth, the existing literature has often used macroeconomic-level (firm-level) data to investigate the relationship between Fintech and economic growth (risk-taking). However, this study uses bank-level data to examine the effects of Fintech on the banking industry. Fifth, this is a valuable case study that examines the relationship between Fintech and efficiency in the Taiwanese banking industry. Taiwan has a comprehensive IT infrastructure, talent pool, and industrial chain. The study examines whether Fintech plays a crucial role in bank efficiency and aims to answer the following question: Can banks improve their efficiency through Fintech?

The remainder of this study is organized as follows. Section 1 contains the introduction. Section 2 provides a literature review on the relationship between fintech and bank efficiency. Section 3 describes the model specification, data source, and empirical design. Section 4 presents the empirical results and discussion. The conclusions are presented in Section 5.

## Background

Similar to the early production of ATMs, credit cards, and telephone banking, Internet banking services have exhibited rapid growth in Taiwan since 2000 (Fintech 2.0). For instance, Bank SinoPac launched its first Internet banking service with the "Money Management Account" (MMA) in 2000. MMA is a popular fintech-related service that attracts many users, increasing the volume of transactions and profits. The Taiwan Network Information Centre report showed an Internet banking utilization rate of 36.7% (Data source: http://www.reporttwnic.tw/2019, Taiwan Network Information Centre). Furthermore, the development of digital payment systems has been very rapid in the past decade (this development is part of Fintech 3.0). According to the Market Intelligence & Consulting Institute's (MIC) 2020 mobile payment consumer survey, mobile payments are 60.3% lower than payments through entity cards (76.3%) and cash (75.5%). For stores that support mobile payment, consumer willingness to use mobile payment has grown from 22.9% to 37.2% since 2018 to 2020 (Data source: https://mic.iii.org.tw/news.aspx?id=593. Market Intelligence & Consulting Institute, Taiwan.).

Moreover, it is expected that IT-based service delivery increases savings, subsequently improving bank efficiency. Mobile payment systems (e.g., banking electronic transaction systems) significantly improve the level of convenience associated with services and decrease bilateral operation costs. It is important to investigate the effect of Fintech on bank efficiency. On the one hand, Taiwan has a healthy IT infrastructure and internationally renowned IT enterprises such as Foxconn, TSMC, Quanta, and Mediatek. However, the international role of the Taiwanese banking industry is not as important as that of the Chinese or Hong Kong banking industries in the Greater China region. On the other hand, banks are not vigorously promoting initiatives related to Fintech 3.0. For example, mobile payment systems were only recently adopted by customers in 2016.

## Literature review and hypotheses

IT productivity has always been a concern in academia and industry. Previous studies have focused on the impact of IT on firm performance using cross-sectional and panel data. For example, Shu and Strassmann [8] found that IT investment is the most significant predictor of product margin. IT generally increases product margins better than labor. Since the late 1960s, initiatives related to financial technology (Fintech 1.0) have been applied to the financial services industry. However, Fintech has just recently come to scholars' attention in the past decade.

Most studies find positive effects of fintech on bank efficiency (e.g., Lee et al., [6]; Cho and Chen, [9]; Dong et al., [10]). It enhances bank efficiency in terms of gross margin, employee productivity, and operational talent management. These studies employ regression analysis and other relatively simpler methods. In some regression studies, fintech-related initiatives were used as independent variables. Other studies used multiple bank efficiency scores, incorporating fintech-related initiatives in econometric modeling. Parametric and non-parametric approaches are popular techniques for measuring bank efficiency. For instance, Cook et al. [4] established a model based on data envelopment analysis (DEA)-based model to measure the effects of e-business activities of e-branches on bank efficiency. However, their results indicated the e-branches' fintech-related initiatives did not show bank efficiency improvements compared to traditional branches' best practices.

Econometric approaches have proven effective for evaluating bank efficiency when multiple bank efficiency scores are analyzed. For emphasis, the operative term in this definition is *bank efficiency*, which implies that producers may use sub-optimal decisions to maximize or minimize objective functions. Beccalli [3] used the Fourier flexible cost function and found that, even when banks are major IT investors, there is little relationship between IT investment and bank efficiency. Lee et al. [6] applied the stochastic metafrontier approach to measure bank efficiency, and they find Fintech innovation improves the banks' cost efficiency. Overall, fintech improves bank efficiency by breaking down geographic barriers, establishing new marketing and sales channels, and improving customer life cycle management. This study evaluates bank efficiency by considering fintech-related initiatives as independent variables in a cost function. Based on these arguments, I propose the following hypotheses:

$H_1$: *Fintech-related initiative affect bank efficiency.*

Generally, fintech refers to the combination of finance and IT, using fintech-related initiatives to improve bank efficiency. For example, banks offer ATMs, credit cards, and internet banking as initiatives related to Fintech 2.0 and mobile payment systems and digital finance products as initiatives related to Fintech 3.0. This study used the deposit as an output variable to measure bank efficiency. ATMs are used by banks to capture or protect deposit market

share in return for higher levels of convenience to their depositors (Wang et al., [11]). For example, it is assumed that customers use ATMs to avoid some unnecessary inconvenience, thereby increasing service delivery speed and decreasing related costs to the consumer.

Humphrey and Berger [12] analyzed payment system costs. Their findings showed that cash was the cheapest payment system from a social cost perspective, followed by point-of-sale and ATM bill payments. Frame and White [13] indicated that financial innovation could refer to new services, improved production processes, or organizational forms of innovation. This finding implies that Fintech positively impacts firm performance. However, Cook et al. [4] highlighted the importance of increasing service delivery speed and significantly decreasing costs through ATMs, telephone banking, and other online and electronic means. Sathye and Sathye [14] found that ATM intensity had a significant negative association with bank efficiency, and ascribed this to banks' investment in Fintech—such as ATMs—and their inability to reduce labor costs, given that many processes continue to be manual. Kondo [15] found that ATMs did not influence banks' return on assets (ROA) in Japan. Fintech provides a variety of new channels, such as internet banking, telephone banking, and ATMs. It contributes to the precision, speed, and confidentiality of customer transactions, while boosting customer convenience (Liao, [16]). However, empirical results are mixed regarding the relationship between ATMs and bank efficiency. Based on these arguments, this study proposes the following hypotheses:

$H_{2a}$: *ATMs have a positive impact on bank efficiency.*

$H_{2b}$: *ATMs have a negative impact on bank efficiency.*

Mobile payment systems have become increasingly popular because of the widespread use of Internet-based shopping and banking. Digital payments and banking have been introduced as new ways to conduct convenient and effective financial transactions (Leong et al., [17]). Mobile payment platforms have significantly increased the level of convenience associated with services and decreased bilateral operation costs. However, banks' investments in fintech-related initiatives tend to be risky without guaranteed results (Cho and Chen, [5]).

Moreover, the tendency for customers to choose cash vs. non-cash payment systems may depend on transaction costs. This situation illustrates the effect of payment systems on bank efficiency. Notably, non-cash payment systems' operating costs are lower than cash payment systems. In summary, previous studies focused on mobile payment systems adoption and its effect on convenience. Few studies have quantified the effects of mobile payment systems on bank efficiency. Thus, this study aims to evaluate the benefits of mobile payment systems on bank efficiency.

Based on the above arguments, this study proposes the following hypotheses:

$H_{3a}$: *Mobile payment systems have a positive impact on bank efficiency.*

$H_{3b}$: *Mobile payment systems have a negative impact on bank efficiency.*

IT investment is the core of fintech development. Thus, it indirectly affects bank efficiency. In this study, I answer the question: to what extent does IT investment impact bank efficiency? While studies have argued that IT investments increase revenue and savings (Mithas et al., [18]), Strassman [19] insisted that it cannot contribute to productivity or corporate performance as expected. It is argued that IT investment has grown remarkably; however, the productivity of firms and the entire economy has been relatively slow. Beccalli [3] found little to no connection between IT investment and bank efficiency, a "profitability paradox." Gupta et al. [20] confirmed this profitability paradox in the Indian banking sector. This phenomenon raises doubts regarding IT investments' positive effects (Rao and Mandal, [21]). On the

contrary, Ou et al. [22] found that IT investment is positively associated with bank efficiency in Taiwan. Regardless, what is clear is that quantitative and qualitative studies on the relationship between IT investment and bank efficiency are sparse. Therefore, this study proposes the following hypotheses:

$H_{4a}$: *IT investment has a positive impact on bank efficiency.*

$H_{4b}$: *IT investment has a negative impact on bank efficiency.*

Overall, this study investigates how fintech impacts bank efficiency in Taiwan. It extends and integrates prior literature by comparing bank efficiency with and without fintech-related initiatives (i.e., e-business or IT outputs). To the best of my knowledge, this study is the first to consider how fintech-related initiatives affect bank efficiency (bank cost, technology, and scale) in Taiwan.

## Methodology

### Bank efficiency with fintech-related initiatives

Numerous studies have considered bank efficiency as a vital topic. Following Bauer et al. [23], this study employs parametric and non-parametric approaches to measure bank efficiency. A parametric approach was developed by Aigner et al. [24]. In their study, they measured bank efficiency in terms of the difference in costs between banks that use traditional best practices vs. non-traditional practices (fintech-related initiatives). The cost function applicable to panel data can be written as follows:

$$TC_{it} = f(y_{it}, w_{it}; \beta) + \varepsilon_{it} \tag{1}$$

where $\varepsilon_{it} = u_{it} + v_{it}$, $i = 1, \cdots. N.$ $t = 1, \cdots, T$.

Here, $TC_{it}$ is bank $i$ of the total cost on year $t$, $y_{it}$ indicates the bank of output, $w_{it}$ indicates the bank of input price, and $\beta$ is the vector of unknown coefficients for the associated output and input price variables in the Eq (1). $u_{it}$ captures unobserved phenomena, while $v_{it}$ captures inefficiency and is independent of $u_{it}$.

As an extension of the stochastic frontier analysis (SFA) and the distribution-free approach (DFA). Battese and Coelli [25] suggested a one-step approach to estimate bank efficiency and simultaneously incorporate exogenous variables that impact bank efficiency. In this model, $v_{it}$ is assumed to have a systematic component $\theta' z_{it}$ (which is correlated with the exogenous variables) and a random component. Thus, by inserting the residual term $v_{it}$ in Eq (1), the cost function can be written as follows:

$$lnTC_{it} = f(y_{it}, w_{it}; \beta) + u_{it} + \theta' z_{it} + e_{it} \tag{2}$$

where Z is a set of variables that affect bank efficiency, the bank efficiency term assumes that it is independent and identically half-normal distribution (See Battese and Coelli [25] for details).

The following is done to measure bank efficiency with fintech-related initiatives. Using an input-oriented DEA model, variable returns to scale (VRS) were calculated for the production plan (x, y) of production unit I. The linear programming function is as follows:

$$C^* = \frac{min}{x} \{w'x|y, \ (y, x) \in PPS\} = \ C^*(y, x) \tag{3}$$

Cost minimization requires that the optimal input x under output y be fixed. This study used VRS cost minimization. Suppose that, in addition to the assumption that cost efficiency is

$C = w^T x^E / w^T x^A$, where w is an input price vector, $C(y,w) = w^T x^E$ is the total cost of efficiency, $x^A$ is an input vector, and input prices of $w \in R_+^N$ are given. Therefore, cost efficiency can be rewritten as follows:

$$min_{z,x_i} \; w_{i0} x_{i0}^*$$

$$\text{s.t. } \sum_{k=1}^{k} z_k y_{jk} - y_{j0} \geq 0, j = 1, 2, \cdots, m.$$

$$\sum_{k=1}^{k} z_k x_{ik} - \lambda x_{i0}^* \leq 0, \; i = 1, 2, \cdots, n. \; \sum_{k=1}^{k} z_k = 1 \tag{4}$$

where $z_k$ is the scaling vector for the production plans, $x_{i0}^*$ is calculated by LP, $x_{i0}^*$ is cost-minimizing vector of input quantities for the evaluated bank, given the input prices $w_{i0}$, the $w_{ik}$ is $i^{th}$ input price of $k^{th}$ bank, and output level $y_{i0}$, and $\lambda$ is the efficiency scale.

The cost efficiency of bank i ($CE_{ik}$) is calculated as the ratio of the bank's estimated minimum cost of producing a certain output to the actual cost of production. Thus, $CE_{ik}$ can be calculated as follows:

$$CE_{ik} = \frac{\sum_{i=1}^{n} w_{ik} x_{ik}^*}{\sum_{i=1}^{n} w_{ik} x_{ik}} \tag{5}$$

The input-output specification of this study is based on the intermediation approach suggested by Wang et al. [11], Bauer et al. [23], and Liao [16], in which salary, physical capital, and fund expenses are the input factors used to produce earnings assets, labor, and interest prices in the basic model. The price of wages ($w_1$) is the total salary of all employees, and the price of funds ($w_2$) is the interest paid to all funding. The price of physical capital ($w_3$) is the physical capital expense of the book value of the fixed asset. The two outputs are total loans ($Y_1$) and total deposits ($Y_2$), which are commonly used in the literature.

Fintech is widely promoted in financial markets and institutional products and services. Few studies have evaluated the effects of fintech-related initiatives on bank efficiency. Wang et al. [11] stated that it is important to evaluate the following to measure bank efficiency: (1) the efficiency of IT-related initiatives and (2) the efficiency of transforming IT-produced intermediate output variables. Thus, this reflects the importance of fintech-related initiatives to bank efficiency. Therefore, fintech-related initiatives, such as self-service technology, ATMs, and non-cash payment systems, are identified. Cook et al. [4] argued the importance of increasing service delivery speed and decreasing costs in significant proportions through ATMs, telephone banking, and other online and electronic means. Sathye and Sathye [14] used DEA to discuss whether an investment in self-service technology, such as ATMs, improves bank efficiency. Existing literature on payment systems has focused on comparing cash with non-cash (credit or debit cards) payment systems. Notably, payment by credit card is motivated by the option to make installment payments (Boden et al., [26]). To increase their operating revenue, most banks proactively allow mobile and credit card payments to attract customers willing to use them.

This study uses the following proxies for Fintech-related initiatives: 1) the value of ATM transactions and 2) the number of transactions by credit card annually, It compares the results of the parametric and non-parametric analysis. An increased volume of ATM transactions and transactions by credit card would lower operating costs, leading to increased earnings and bank efficiency.

## Relationship between bank efficiency and fintech-related initiatives

Based on the aforementioned discussion and hypotheses, this study investigates the determinants of bank efficiency. The baseline equation is as follows:

$$BEFF_{it} = \partial_0 + \partial_1 ATM_{it} + \partial_2 ITE_{it} + \partial_3 MPAY_{it} + \partial_4 TPAY_{it} +$$
$$\partial_5 SIZE_{it} + \partial_6 NPL_{it} + \partial_7 EA_{it} + \partial_8 MANAGE_{it} + W_{it}$$

(6)

where $BEFF_{it}$ indicates bank managerial efficiency (X-efficiency; from the DFA) bank inefficiency (from the SFA), and bank cost efficiency (from the DEA); ATM indicates the number of ATMs; ITE indicates annual IT investment; MPAY is a dummy variable whose value is 1 if the bank cooperated with Taiwan Mobile Payment Co. Ltd. (TWMP, a mobile payment systems developer since 2014) and 0 otherwise; TPAY is a dummy variable whose value is 1 if the bank used TWMP's payment app system and 0 otherwise; SIZE indicates the natural logarithm of bank assets; NPL indicates non-performing loans; EA indicates equity-assets ratio; and MANAGE indicates total operating revenue-total operating cost.

Fintech refers to business models, technological applications, operating processes, and innovative products that promote financial innovation through IT and have a significant impact on financial services (Financial Stability Board, [27]). This study uses several variables to proxy initiatives related to Fintech 2.0 and Fintech 3.0 in regular transactions. A proxy for initiatives related to Fintech 2.0 is ATM (this is a variable name, not to be confused with "Automated Teller Machine"), measured by the logarithm of the number of ATMs of each bank. Le and Ngo [28] stated that IT-based methods of service and product could improve bank efficiency. When customers go to a teller to cash a check, they wait in the queue. However, the waiting time can be shorter than the amount of time needed to withdraw an equally large amount from an ATM (Cook et al., [4]).

A proxy for initiatives related to Fintech 3.0 is mobile payment systems. Banks serve as intermediary institutions in Taiwan in a sense that customers prefer to use credit cards in traditional banking systems. Innovations in mobile payment systems have recently introduced several e-payment channels. Consequently, fintech companies have been established.

In 2014, the government established the TWMP to help the financial industry launch mobile payment systems and promote e-commerce. The TWMP provides host card emulation and tokenization payment platform services for financial institutions, reducing the cost of mobile payment systems. Following Berger et al. [29], this study uses selection indicator dummy variables. MPAY is a dummy variable whose value is 1 if the bank cooperated with TWMP and 0 otherwise. In this study, selection indicator dummies variables defined all dummy variables. For example, MPAY as a dummy variable in each dummy variable equal to zero before the bank did not cooperate with TWMP; when the bank started the year's collaboration with TWMP dummy variable was equal to one. Furthermore, TWMP's payment system was used by customers to make deals, pay taxes, and transfer money. It is a tokenization payment platform for financial institutions that began in 2017. Thus, TPAY is a dummy variable whose value is 1 if the bank used TWMP's payment service and 0 otherwise.

A proxy for IT investment is called ITE. It refers to the total operating cost of hardware, software, and other IT service expenses. Mithas et al. [18] showed that IT investment positively affects firm revenue. They showed that IT investment decreases overall operating expenses. However, Beccalli [3] argued that there is an insignificant relationship between IT investment and bank efficiency.

For this study, data collection was facilitated using the following sources: (1) *Taiwan Economics Journal* (TEJ) database (the primary financial report data source); (2) annual bank reports (for supplementary financial variables); and (3) banks' websites (for variables related to

**Table 1. Descriptive statistics of bank characteristics.**

| Variables | Mean | Std. | Max. | Min. |
|---|---|---|---|---|
| *Cost Function* | | | | |
| Loans | 738613.73 | 287591.21 | 2869204.52 | 25614.58 |
| Deposits | 990791.71 | 4651432.17 | 4172738.84 | 34502.57 |
| Value of transaction for ATM | 248.26 | 213.44 | 2683.12 | 4.426 |
| Transaction amount by Credit card | 65.381 | 103.61 | 543.275 | 0.559 |
| Salary | 5805.15 | 2187.92 | 32050.4 | 102.44 |
| physical capital expense | 9738.62 | 11282.19 | 63192.82 | 250.29 |
| Interest expense | 5573.61 | 6940.11 | 30884.89 | 135.738 |
| Price of wage | 1201.31 | 393.53 | 2671.26 | 49.37 |
| Price of physical capital | 0.5246 | 0.3564 | 3.2602 | 0.0391 |
| Price of funds | 0.0111 | 0.011 | 0.0447 | 0.0013 |
| *Regression Variables* | | | | |
| ATM | 5.8375 | 0.1567 | 8.776 | 3.688 |
| ITE | 0.0267 | 0.0093 | 0.1694 | 0 |
| TPAY | 0.0982 | 0.707 | 1 | 0 |
| MPAY | 0.203 | 0.7071 | 1 | 17.5220 |
| SIZE | 20.424 | 0.2372 | 22.409 | 0.01 |
| NPL | 0.6981 | 0.792 | 7.64 | 3.396 |
| EA | 6.929 | 1.208 | 21.767 | 3.396 |
| MANAGE | 1.7958 | 0.5948 | 4.8047 | -4.64 |

Notes: the primary data source for this study was the Taiwan Economics Journal (TEJ). The input-output variables unit measured by New Taiwan Dollars (NTD) millions, price of funds and physical capital measured by percent.

IT investment). The unbalanced panel data includes data from 2007 to 2020 on 32 banks. The total number of observations is 448. Table 1 provides the descriptive statistics of the variables.

## Empirical results

### Evaluation of bank efficiency

This section reports the average bank efficiency scores estimated using SFA, DFA, and DEA models. The results of parametric estimation (without fintech-related initiatives) are presented in Table 2. The mean X-efficiency scores are 0.7849 and 0.9448, implying that the average bank could reduce its costs by 21.51% and 5.52%, respectively, by adopting the best practice in Taiwan (To conserve space, only the transaction values of ATM proxy fintech-related initiatives were shown because their results are almost the same and available upon request.). This shows that bank efficiency with fintech-related initiatives (proposed model) is better than without fintech-related initiatives (traditional model), which is consistent with Ou et al. [22] but inconsistent with Sathye and Sathye [14]. The efficiency score increased from 0.942 in 2008 to 0.9461 in 2019 with fintech-related initiatives. This means that initiatives related to Fintech will function to improve bank efficiency until the Fintech application has matured. The SFA model showed the same results wherein the inefficiency score in the proposed model is lower than the traditional model.

Table 2 also presents the results of bank efficiency using the DEA model. The cost efficiency score with fintech-related initiatives (proposed model) is 0.909. This is higher than that without fintech-related initiatives (traditional model), whose cost efficiency score is 0.871. On

**Table 2. Results of banks efficiency by parametric and non-parametric approaches.**

| Years | DFA | | SFA | | TE | | AE | | CE | |
|---|---|---|---|---|---|---|---|---|---|---|
| | $XEFF_{Basic}$ | $XEFF_{FT}$ | $INEFF_{Basic}$ | $INEFF_{FT}$ | $TE_{Basic}$ | $TE_{FT}$ | $AE_{Basic}$ | $AE_{FT}$ | $CE_{Basic}$ | $CE_{FT}$ |
| 2007 | 0.623 | 0.942 | 0.6093 | 0.471 | 0.489 | 0.289 | 0.712 | 0.312 | 0.644 | 0.903 |
| 2008 | 0.464 | 0.943 | 0.6118 | 0.461 | 0.696 | 0.714 | 0.790 | 0.795 | 0.876 | 0.895 |
| 2009 | 0.788 | 0.941 | 0.6269 | 0.476 | 0.674 | 0.744 | 0.781 | 0.839 | 0.865 | 0.885 |
| 2010 | 0.821 | 0.942 | 0.6332 | 0.464 | 0.689 | 0.788 | 0.772 | 0.854 | 0.893 | 0.920 |
| 2011 | 0.820 | 0.944 | 0.6254 | 0.465 | 0.639 | 0.730 | 0.703 | 0.783 | 0.903 | 0.925 |
| 2012 | 0.829 | 0.945 | 0.624 | 0.471 | 0.595 | 0.690 | 0.669 | 0.753 | 0.879 | 0.905 |
| 2013 | 0.784 | 0.946 | 0.6175 | 0.474 | 0.671 | 0.720 | 0.750 | 0.790 | 0.889 | 0.908 |
| 2014 | 0.795 | 0.947 | 0.6005 | 0.465 | 0.645 | 0.715 | 0.720 | 0.778 | 0.893 | 0.915 |
| 2015 | 0.826 | 0.946 | 0.5846 | 0.452 | 0.632 | 0.715 | 0.713 | 0.788 | 0.887 | 0.903 |
| 2016 | 0.831 | 0.946 | 0.5794 | 0.45 | 0.641 | 0.702 | 0.711 | 0.767 | 0.899 | 0.908 |
| 2017 | 0.892 | 0.947 | 0.5741 | 0.447 | 0.612 | 0.695 | 0.678 | 0.757 | 0.900 | 0.912 |
| 2018 | 0.841 | 0.946 | 0.5703 | 0.445 | 0.611 | 0.699 | 0.689 | 0.760 | 0.890 | 0.919 |
| 2019 | 0.817 | 0.947 | 0.5656 | 0.443 | 0.624 | 0.695 | 0.710 | 0.761 | 0.885 | 0.912 |
| 2020 | 0.858 | 0.945 | 0.554 | 0.425 | 0.612 | 0.688 | 0.696 | 0.748 | 0.883 | 0.921 |
| mean | 0.785 | 0.945 | 0.5983 | 0.4579 | 0.631 | 0.685 | 0.721 | 0.749 | 0.871 | 0.909 |

Notes: $XEFF_{Basic}$ and $XEFF_{FT}$ indicates banks efficiency measured by DFA model, and $INEFF_{Basic}$ and $INEFF_{FT}$ indicates banks efficiency measured by SFA. Cost efficiency (CE), allocative efficiency (AE) and technical efficiency (TE) indicate bank efficiency measured by DEA. *Basic* is the efficiency value evaluated by the traditional model, and *FT* is the efficiency value evaluated by considering fintech-related initiatives.

average, approximately 10% of banks' costs were wasted relative to the best-practice banks within the sample, producing the same output and facing the same conditions. Bank efficiency was measured by comparing the results in the proposed and traditional models. Banks that used fintech-related initiatives saved approximately 3.8% of their total operating costs. However, this is very low. The results of the parametric and non-parametric models were similar. Initiatives related to Fintech 2.0 increased bank efficiency.

## Results of sensitive analysis

This section explains the results of the SFA, DFA, and DEA models. These models were used to quantify the relationship between bank efficiency and fintech-related initiatives. Bank efficiency was modeled against proxy variables for fintech-related initiatives. In the traditional model, fintech-related initiatives are non-existent. In the proposed model, fintech-related initiatives are existent. As seen in Table 3, the efficiency scores with fintech-related initiatives were significantly higher in the SFA, DFA, and DEA models. Thus, fintech affects bank efficiency, supporting $H_1$.

Moreover, two kinds of mobile payment systems are compared to check whether bank efficiency has improved. The results of the parametric analysis show that mobile payment systems did not significantly improve bank efficiency. However, bank efficiency improved when the TWMP payment system was used, as shown in the DEA model. Thus, the impact of mobile payment systems on bank efficiency remains ambiguous.

Tallon [30] stated that SIZE might be a key determinant of business strategy. In the context of service delivery, small and large banks may have different approaches in fintech. While fintech is more important for service delivery in large banks, the indirect effects of fintech may be higher for those in small banks. Furthermore, the mediating effect of organizational innovation on the relationship between e-business and bank efficiency is analyzed (Soto-Acosta,

**Table 3. Comparison of bank efficiency among different models.**

| | Efficiency score | T-test | One-way ANOVA |
|---|---|---|---|
| **Parametric approach** | | | |
| $INEFF_{Basic}$ vs. $INEFF_{FT}$ | 0.5982<br>0.458 | -8.248*** | 68.028*** |
| $XEFF_{Basic}$ vs. $XEFF_{FT}$ | 0.785<br>0.945 | 25.117*** | 630.86*** |
| **Non-Parametric approach (DEA)** | | | |
| $CE_{Basic}$ vs. $CE_{FT}$ | 0.8705<br>0.9094 | 4.016*** | 16.127*** |
| $AE_{Basic}$ vs. $AE_{FT}$ | 0.7211<br>0.7489 | 2.039** | 4.157*** |
| $TE_{Basic}$ vs. $TE_{FT}$ | 0.631<br>0.6845 | 3.59*** | 12.886*** |

Notes: This study test whether has a gap with three models, as follows T test, one-way ANOVE test with F-statistics.

* Significant level at the $\alpha = 0.1$,

**at $\alpha = 0.05$ and

***at $\alpha = 0.01$.

[31]). Thus, this study classifies the sample of banks into large and small to evaluate whether large banks perform better than small banks. The sample is classified into two groups, namely, large banks and small banks, in which total assets of more than 100 billion new Taiwan dollars (NTD) is considered a large bank or a small bank otherwise. The results show that small banks' efficiency is higher than that of large banks (Table 4). This implies that small banks are relatively efficient at managing their total cost-to-total outputs ratio, including fintech-related initiatives.

Subsequently, this study tests whether an organization has a significant effect on bank efficiency. This study investigates whether banks joining financial holding companies (FHCs) can promote their bank efficiency better through IT investment. Our findings show that bank efficiency does not improve with FHC membership. One possible reason for this phenomenon is that larger banks are less inclined to use fintech to improve brank efficiency. Overall, these results strongly support $H_1$.

## Determination on bank efficiency

In this section, I explain the results of the regression analysis applied to panel data. We regressed bank efficiency on the proxy variables for the initiatives related to Fintech 2.0 and Fintech 3.0. As shown in Table 5, all variables of Variance Inflation Factor (VIF) were less than 10, implying no collinearity problems in the regression analysis. The coefficient of ATM is negative and significant in most of the models, supporting $H_{2b}$. This suggests that ATMs do not increase bank efficiency. One possible reason for this phenomenon is that transactions under NTD $500 are free of charge in Taiwan; however, there are essential costs in expanding an ATM network (Liao, [16]). Thus, more ATM transactions do not necessarily lead to increased bank efficiency. A possible alternative reason is that ATMs are ineffective payment systems. Customers are still used to the traditional payment or transfer modes (Tsai et al., [32]).

The coefficients of MPAY and TPAY are significantly positive (see the $XEFF_{FT}$ column). However, their signs are insignificant in the other columns. Moreover, mobile payment systems are positively associated with bank efficiency. Thus, there is slight evidence to support

**Table 4. Results of sensitive analysis.**

|  | Efficiency score | Obs. | T-test | One-way ANOVA |
|---|---|---|---|---|
| **TPAY** |  |  |  |  |
| XEFF | 0.739 | 154 | -5.484*** | 30.071*** |
|  | 0.809 | 294 |  |  |
| CE | 0.974 | 154 | 8.61*** | 74.136*** |
|  | 0.826 | 294 |  |  |
| **MPAY** |  |  |  |  |
| XEFF | 0.736 | 182 | -6.681*** | 44.637*** |
|  | 0.818 | 266 |  |  |
| CE | 0.899 | 182 | 3.096*** | 9.585*** |
|  | 0.851 | 266 |  |  |
| **FHC** |  |  |  |  |
| XEFF | 0.766 | 210 | -2.809*** | 7.889*** |
|  | 0.801 | 238 |  |  |
| CE | 0.881 | 210 | 1.344 | 1.806 |
|  | 0.861 | 238 |  |  |
| **SIZE** |  |  |  |  |
| XEFF | 0.753 | 223 | -5.192*** | 26.954*** |
|  | 0.816 | 225 |  |  |
| CE | 0.913 | 223 | 5.821*** | 33.89*** |
|  | 0.828 | 225 |  |  |

Notes:

* Significant level at the $\alpha = 0.1$,

**at $\alpha = 0.05$ and

***at $\alpha = 0.01$.

$H_{3a}$ and strong evidence to reject $H_{3b}$. This implies that mobile payment systems increase the frequency of banking transactions but do not necessarily increase bank efficiency. Therefore, although mobile payments significantly increase the level of convenience associated with services and decrease bilateral operation costs, these transactions are accompanied by significantly higher operating costs to the bank. Fintech may bring more harm than good to traditional banks in the early stages. However, with the weakening of the competitive effect, the demonstration effect gradually becomes prominent and improves financial efficiency (Wu et al., [33]).

The coefficient of ITE is significantly positive in the $XEFF_{Basic}$ column. However, its signs are negative or insignificant in the other columns. These results are consistent with those of Beccalli [3] but inconsistent with Mithas et al. [18] and Lin [34]. This study finds slight evidence for the relationship between IT investment and bank efficiency, indicating an IT "productivity paradox" in Taiwan's banking industry. The IT Productivity Paradox (or the Solow paradox) indicates that productivity growth may not increase despite sizeable investment in IT (Shu and Strassmann, [8]). These results provide weak support for $H_{4a}$, implying that banks' inability to maximize the benefits of IT investment does not directly contribute to bank efficiency. This suggests that the adoption of increasingly expensive IT infrastructure has become a structural component of competition in the banking industry (Beccalli, [3]).

The coefficient of SIZE is significantly positive in some columns but insignificant in some, consistent with Liao [35]. Thus, the implications are ambiguous. These findings indicate that big banks are unsure about being more efficient or profitable. Thus, big banks should better understand their resources can be effectively utilized, and are concerned with the optimal

**Table 5. Results of determinant bank efficiency with fintech.**

| | $XEFF_{FT}$ | | $XEFF_{Basic}$ | | $CE_{FT}$ | | $CE_{Basic}$ | | |
|---|---|---|---|---|---|---|---|---|---|
| | FM | RM | FM | RM | FM | RM | FM | RM | VIF |
| **ATM** | -0.003 (-6.062)*** | -0.006 (-14.25)*** | -0.049 (-2.336)** | -0.054 (-4.8)*** | 0.031 (2.848)*** | 0.040 (3.024)*** | 0.012 (0.831) | 0.012 (0.741) | **3.328** |
| **ITE** | -0.0004 (-0.071) | 0.028 (4.546)*** | 0.721 (3.629)*** | 0.578 (2.705)*** | -0.054 (-0.685) | -0.465 (-2.41)** | 0.046 (0.308) | 0.009 (0.034) | **1.076** |
| **MPAY** | 0.002 (7.808)*** | 0.004 (9.000)*** | 0.004 (0.290) | 0.010 (0.690) | -0.004 (-1.227) | -0.016 (-1.213) | -0.016 (-2.186)** | -0.006 (-0.342) | **1.31** |
| **TPAY** | 0.001 (4.127)*** | 0.002 (3.58)*** | -0.005 (0.289) | 0.004 (0.221) | 0.0004 (0.131) | 0.008 (0.510) | -0.006 (-0.868) | -0.017 (-0.843) | **1.442** |
| **SIZE** | -0.006 (-15.312)*** | -0.013 (-30.71)*** | 0.092 (4.448)*** | 0.037 (3.018)*** | -0.0006 (-0.069) | -0.004 (-0.276) | -0.006 (-0.398) | -0.005 (-0.292) | **3.944** |
| **NPL** | -0.002 (-14.729)*** | -0.003 (-17.37)*** | -0.090 (-13.494)*** | -0.083 (-13.2)*** | -0.005 (-1.902)* | -0.010 (-1.796)* | -0.037 (-6.322)*** | -0.062 (-8.3)*** | **1.307** |
| **EA** | 0.001 (17.875)*** | 0.001 (13.344)*** | -0.001 (-0.383) | 0.003 (0.905) | 0.0007 (0.417) | 0.004 (1.291) | -0.0002 (-0.095) | -0.003 (-0.813) | **1.203** |
| **MANAGE** | -0.0005 (-2.143)*** | -0.0007 (-2.564)** | 0.011 (1.010) | 0.004 (0.381) | -0.0001 (-0.029) | -0.005 (-0.566) | 0.028 (2.963)*** | 0.029 (2.388)** | **1.426** |
| **Obs.** | 448 | 448 | 448 | 448 | 448 | 448 | 448 | 448 | |
| **$R^2$** | 0.994 | 0.825 | 0.647 | 0.428 | 0.789 | 0.049 | 0.831 | 0.207 | |
| **Hausman test Chi-square** | 325.442 (P-value = 0.000) | | 28.834 (P-value = 0.0003) | | 32.339 (P-value = 0.0001) | | 21.106 (P-value = 0.0069) | | |

Notes: ATM indicates the number of automated teller machines, ITE indicates IT expenses per year, MPAY is a dummy variable that equals one if the bank cooperation with Taiwan mobile payment platform and zero otherwise, TPAY is a dummy variable that equals one if the bank cooperation with Taiwan-pay app service and zero otherwise; SIZE indicates the natural logarithm of bank assets, and NPL indicates non-performing loans. EA indicates equity-assets ratio and MANAGE indicates total operating revenue-total operating cost.

\* Significant level at the α = 0.1,

\*\*at α = 0.05 and

\*\*\*at α = 0.01.

distribution of products and services. The coefficient of NPL is negatively significant across all columns, which is consistent with the existing literature. The coefficient of EA is positively significant in the $XEFF_{FT}$ column, but insignificant in the other columns. This implies that a higher EA does not necessarily improve bank efficiency. The sign of the coefficient of MANAGE is insignificant across most columns, implying that the revenue-cost ratio also does not improve bank efficiency.

Overall, these results show that initiatives related to Fintech 2.0 do not have a positive effect on bank efficiency in Taiwan. However, the initiatives related to Fintech 3.0 slightly improve bank efficiency. One possible reason for this phenomenon is that fintech in new businesses is perceived as highly risky, with higher R&D costs.

## Additional test: SFA model

This section investigates the determinants of bank inefficiency by the SFA model, which involves specifying a regression model for cost inefficiency effects. As shown in Table 6, the coefficient of ATM is significantly positive in the FT model, implying that the extension of ATM networks by a bank does not improve bank efficiency. Previous studies have shown ambiguous results regarding the effects of ATMs on bank performance. Le and Ngo [28] and Ou et al. [22] show that ATM network are associated with positive cost efficiency. However,

**Table 6. Results of determinant bank inefficiency with SFA model.**

| | Two-stage SFA model | |
|:---:|:---:|:---:|
| | **Basic model** | **FT model** |
| *ATM* | 0.1497 (8.82)*** | 0.24 (6.477)*** |
| *ITE* | 0.2125 (0.623) | -0.071 (-0.098) |
| *TPAY* | 0.045 (1.355) | -0.471 (-2.605)** |
| *MPAY* | 0.0398 (1.483) | -0.174 (-2.392)** |
| *SIZE* | 0.463 (15.29)*** | -0.176 (-3.368)*** |
| *NPL* | 0.0954 (4.041)*** | 0.0721 (2.421)** |
| *EA* | 0.0288 (2.168)** | 0.0253 (0.635) |
| *MANAGE* | -0.055 (-2.281)** | -0.085 (-1.259) |
| **Obs**. | 448 | 448 |
| Log-likelihood function | 263.42 | 122.249 |
| LR test of one-side error | 328.83 | 129.534 |

Notes: ATM indicates the number of automated teller machines, ITE indicates IT expenses per year, MPAY is a dummy variable that equals one if the bank cooperation with Taiwan mobile payment platform and zero otherwise, TPAY is a dummy variable that equals one if the bank cooperation with Taiwan-pay app service and zero otherwise; SIZE indicates the natural logarithm of bank assets, and NPL indicates non-performing loans. EA indicates equity-assets ratio and MANAGE indicates total operating revenue-total operating cost.

* Significant level at the $\alpha = 0.1$,

**at $\alpha = 0.05$ and

***at $\alpha = 0.01$.

Kondo [15] found no relationship between ATMs and profitability in Japan. Thus, these results support Hypothesis $H_{2b}$.

The coefficient of ITE is insignificant in all the models, implying that IT investment does not decrease bank inefficiency. These results are consistent with Shu and Strassmann [8] and Mallick and Ho [36] and suggest that Fintech may change work processes and organizational structures but does not change cost efficiency in banks. The coefficient of TPAY and MPAY are significantly negative in the basic model, but insignificant in the FT model. This implies that Fintech 3.0 products (services) have positively affected bank efficiency more than Fintech 2.0 products (services). In sum, these findings are similar to my main results. The SFA model results support the idea that fintech innovation can help reduce bank inefficiency.

## Robustness test

Robustness tests were conducted using different estimation methods, such as the two-stage least squares (2SLS) and panel Granger causality test. This study also tested the alternative definitions of the proxy variables for Fintech 2.0 and Fintech 3.0 and bank performance (e.g., ROA). The specific steps are outlined below.

First, I solve the endogeneity problem using two-stage least squares (2SLS), following Banna et al. [37], who used a lagged model (lagged by one period). As Table 8 shows, these results are similar to the results of the fixed and random effects model applied to panel data.

**Table 7. Results of robustness test.**

|  | XEFF | XEFFB | ROA | | XEFF | | XEFF | XEFFB |
|---|---|---|---|---|---|---|---|---|
|  | 2SLS | 2SLS | FM | RM | FM | RM | OLS | OLS |
| *ATM* | -0.002 (-6.13)*** | -0.038 (4.575)*** | 0.080 (2.352)** | 0.053 (1.163) | -0.004 (-9.025)*** | -0.006 (-15.404)*** | -0.002 (-5.63)*** | -0.025 (-3.052)*** |
| *ITE* | 0.036 (3.248)*** | 0.643 (2.566)** | -0.347 (-0.628) | 0.228 (0.250) | -0.001 (-0.153) | 0.029 (4.577)*** | 0.032 (3.466)*** | 0.485 (2.089)** |
| *TPAY* | 0.002 (2.308)** | -0.011 (-0.473) | -0.067 (-2.356)** | -0.080 (-1.048) |  |  |  |  |
| *MPAY* | 0.004 (5.128)*** | -0.008 (-0.470) | 0.062 (2.25)** | 0.047 (0.734) |  |  |  |  |
| *BLC* |  |  |  |  | 0.002 (6.273)*** | 0.002 (4.416)*** |  |  |
| *TELEP* |  |  |  |  | -0.0001 (-0.199) | 0.001 (1.596) |  |  |
| *TCD* |  |  |  |  |  |  | 0.001 (0.393) | -0.277 (-4.045)*** |
| *TAM* |  |  |  |  |  |  | -0.0002 (-0.146) | 0.099 (3.693)*** |
| *SIZE* | -0.019 (-44.8)*** | 0.015 (1.522) | -0.093 (-2.172)** | -0.108 (-2.121)** | -0.005 (-13.96)*** | -0.012 (-28.126) | -0.019 (-51.9)*** | 0.010 (1.027) |
| *NPL* | -0.007 (-13.1)*** | -0.122 (-10.3)*** | -0.278 (-15.99)*** | -0.321 (-11.8)*** | -0.002 (-14.066)*** | -0.003 (-17.080) | -0.002 (-2.298)** | -0.010 (-0.397) |
| *EA* | 0.001 (7.38)*** | 0.003 (0.824) | -0.030 (-3.88)*** | -0.032 (-2.385)** | 0.002 (20.978)*** | 0.002 (15.499) | 0.0003 (1.845)* | 0.001 (0.396) |
| *MANAGE* | -0.002 (-2.78)*** | -0.009 (-0.598) | 0.803 (26.31)*** | 0.841 (19.543)*** | -0.0009 (-3.518)*** | -0.001 (-3.881) | -0.0004 (-0.797) | 0.036 (2.722)*** |
| **Obs.** | 416 | 416 | 448 | 448 | 448 | 448 | 128 | 128 |
| $R^2$ | 0.961 | 0.337 | 0.861 | 0.645 | 0.994 | 0.805 | 0.990 | 0.342 |
| **Hausman test Chi-square** |  |  | 35.323 (P-value = 0.000) |  | 367.932 (P-value = 0.000) |  |  |  |

Note:TCD indicates the number of Taiwan-pay users; TAM indicates the transaction amounts of Taiwan-pay. TELEP is a dummy variable that equals one if the bank offers telephone banking service and zero otherwise. BLC is a dummy variable that equals one if banks offer financial blockchain correspondence service. Other variables are the same in Table 5 definition.

* Significant level at the α = 0.1,

**at α = 0.05 and

***at α = 0.01.

Second, to further test the impact of initiatives related to Fintech 2.0 and Fintech 3.0 on bank efficiency, I used telephone banking services (TELEP) as a proxy variable for initiatives related to Fintech 2.0. In 2016, Financial Information Services Co. Ltd. collaborated with financial institutions in Taiwan to actively explore the feasibility of applying blockchain technology to financial services. It provides financial information inquiry, protection traceability, and property title documents. I used financial blockchain correspondence services (BLC) as a proxy variable for initiatives related to Fintech 3.0. Like MPAY and TAPY indicators, this study defines BLC and TELEP as both used the selection indicator dummy variable The Eq (6) is re-estimated accordingly. As Table 7 shows, BLC has a significantly positive impact on bank efficiency. Meanwhile, TELEP does not. Following Banna et al. [37], I defined BLC service as a new type of service in Industry 4.0. Accordingly, the impact of post-industrial revolution eras (IR 4.0) on bank efficiency (since 2016) was also investigated. This result is consistent with the

**Table 8. Results of pairwise granger-causality test.**

| Null Hypothesis | Obs. | F-value | P-value |
|---|---|---|---|
| *DFA approach* | | | |
| *ITE does not Granger cause XEFF* | 384 | 2.709* | 0.0679 |
| *XEFF does not Granger cause ITE* | 384 | 2.6317* | 0.0733 |
| *ATM does not Granger cause XEFF* | 384 | 2.5762* | 0.077 |
| *XEFF does not Granger cause ATM* | 384 | 0.5836 | 0.5584 |
| *DEA approach* | | | |
| *ITE does not Granger cause CE* | 384 | 1.174 | 0.3104 |
| *CE does not Granger cause ITE* | 384 | 2.396* | 0.0925 |
| *ATM does not Granger cause CE* | 384 | 0.198 | 0.8207 |
| *CE does not Granger cause ATM* | 384 | 3.544** | 0.0298 |

Notes: XEFF indicates bank X-efficiency by DFA, CE indicates bank cost efficiency by DEA. ATM indicates number of automated teller machines, ITE indicates information technology expense per year. Lags = 2.

* Significant level at the $\alpha = 0.1$,

**at $\alpha = 0.05$ and

***at $\alpha = 0.01$.

former results. This implies that Fintech 3.0-related initiatives functions to improve bank efficiency more than Fintech 2.0-related initiatives.

Third, given the unavailability of data on mobile payment systems, partial information was used to test the relationship between bank efficiency and the number of mobile payment transactions (a proxy variable for mobile payment systems). Thus, the number of TWMP users (TCD) and amount of transactions on TWMP (TAM) in 2017–2020 were used as proxy variables for the construct initiatives related to Fintech 3.0. As Table 8 (Column 8) shows, the coefficient of TAM is positive and significant. However, the coefficients of the other variables are not. This finding is consistent with previous results.

Fourth, this study applied the Granger causality test on panel data to investigate the relationship between fintech-related initiatives and bank efficiency. A unidirectional causality exists between cost efficiency and both IT investment and self-service technology (Table 8). A bidirectional causality exists between X-efficiency and IT investment. These results confirm our finding that there is a significant relationship between bank efficiency and IT investment. However, bank efficiency and self-service technology has a weak relationship.

Finally, a robustness test is conducted using the following as proxy variables of bank efficiency: ROA, return on equity, and others. The results of the regression analysis are consistent, thus confirming the previous findings that mobile payment systems positively affect bank efficiency. However, it also found a positive relationship between ATM and bank efficiency, supporting $H_{2a}$.

## Discussion

Fintech refers to business models, technology applications, operating processes and innovative products that promote financial innovation through information technical means and have a significant impact on financial services. (Financial Stability Board, [27]). This study used several variables to represent fintech-related initiatives for regular transaction deals, such as ATMs, telephone banking and mobile payment and blockchain correspondence service. IT investment also has a pivotal role in evaluating the impact of fintech on bank efficiency.

This study aims to extend the knowledge on how fintech impacts bank efficiency, making four contributions to the literature. First, it remains doubtful whether and how fintech affects a broader set of operational performances over time. To the best of my knowledge, this is the first study to attempt to extend the literature on evaluating how fintech affects bank efficiency using parametric and non-parametric approaches. It takes a more comprehensive perspective and explores the impact mechanism based on the three dimensions of efficiency.

Second, fintech is a series of financial technology providers that enables stretch and better services for banks, businesses, to even individual users. The fintech revolution process has made the world more inclusive and convenient. This study highlights this as the first study to attempt to explain the different impacts of Fintech 2.0, and Fintech 3.0 on bank efficiency, which is Fintech 2.0 products are ATMs and telephone banking, Fintech 3.0 products are mobile payment and blockchain services.

Third, information asymmetry and transaction costs are reduced due to the rapid development of Fintech and increasing IT investment, which contributes to the expansion of the financial market and reconstruction of the financial system. Tanriverdi [38] states that future studies need to explain antecedents such as IT investment. The effect of IT investment must be examined as one of the critical antecedents of Fintech relatedness. This study assesses the magnitude and direction of the Panel data relationship between IT investment and bank efficiency.

Fourth, the 2021 Global Payment Report by Worldpay announced that the average mobile payment usage rates in Internet stores and physical stores are 44.5% and 25.7%, respectively (https://worldpay.globalpaymentsreport.com/en). In comparison, the Institute for Information Industry Survey showed that the first chosen pay form was mobile payment, up to 60.3% of Taiwanese customers in 2020. Greater China areas are commonly characterized by speedily growing economic strength and the evolution of their IT and fintech in the past two decades. For instance, each of the two leading mobile payment service providers, Alipay and WeChat Pay, has around one billion active users in China (Klein, [39]). Studies have responded by analyzing data from emerging Asian countries. This study adds to the literature on the impact of fintech on bank efficiency outside advanced economies, such as Taiwan.

This finding shows that fintech-related initiatives are determinants of bank efficiency; it positively impacts bank efficiency. These results show that initiatives related to Fintech 2.0 do not have a positive effect on bank efficiency in Taiwan. However, the initiatives related to Fintech 3.0 slightly improve bank efficiency. One possible reason for this phenomenon is that fintech in new businesses is perceived as highly risky, with higher R&D costs in the banking industry in Taiwan. These findings provide slight evidence of the relationship between IT investment and bank efficiency, where IT investments improve bank efficiency. However, its contribution to bank efficiency is negligible. One possible reason for this condition is the delayed effect of IT investment in improving bank efficiency, since increasingly expensive IT infrastructures raise operating costs. These results show a weakly improved impact on bank efficiency when fintech is implemented in Taiwanese banks.

## Conclusions

Fintech has become an essential area that needs careful management. However, literature on its impact on bank efficiency is still sparse. Notably, the determinants of bank efficiency have been widely investigated. Few studies have considered Fintech-related initiatives in the context of bank efficiency, with some evaluating Fintech initiatives using linear regression. This is the first study to consider Fintech-related initiatives in connection to bank efficiency in Taiwan,

which assumes Fintech as product output. In additional, the best of my knowledge, this study is the first to explain the difference impacts of Fintech 2.0 and Fintech 3.0 on bank efficiency.

This study addresses this issue using parametric and non-parametric models to measure bank efficiency and its relationship with Fintech. The parametric and non-parametric estimation results are the same. The proposed model measured bank efficiency with fintech-related initiatives. The basic model measured bank efficiency without Fintech-related initiatives. Bank efficiency was significantly higher in the proposed model, supporting $H_1$. Therefore, fintech-related initiatives are determinants of bank efficiency; it positively impacts bank efficiency. These results show that initiatives related to Fintech 2.0 do not have a positive effect on bank efficiency in Taiwan (supporting $H_{2b}$). However, the initiatives related to Fintech 3.0 slightly improve bank efficiency (weakly support $H_{3a}$, but strongly reject $H_{3b}$). Finally, IT investments can increase bank efficiency, but their contribution is negligible (weak support $H_{4a}$). In summary, Fintech 3.0 improves bank efficiency significantly more than Fintech 2.0, and the efficiency of bank IT investment still needs improvement.

In this study, I answer the following question: how can fintech be used to improve bank efficiency? This study has several policy implications. First, from a policy perspective, the government should provide more tax advantages, encourage banks to enlarge IT investment, and allow more non-banking Fintech enterprises to enter the financial market. This would benefit Fintech development and indirectly improve bank efficiency. However, appropriate policy initiatives (e.g., privacy and intellectual property laws) need to be considered. Second, from a managerial perspective, banks investing heavily in Fintech do not necessarily have efficient production processes. They may differ from efficient banks in ways that cannot be rectified by merely increasing their IT investments. Thus, excellent management is crucial. Managers should focus on improving Fintech planning, design, delivery, and strategies. Furthermore, managers should pay attention to the Fintech input–output relationship: achieving the most significant amount of practical work using as little IT investment as possible.

This study also has some limitations, primarily owing to limitations on data accessibility, making analysis difficult, and ultimately impacting the comprehensiveness of the study model. Given the data availability issues, this study used a select dummy variable model and mobile payment data as proxy Fintech 3.0. However, to accurately quantify Fintech 3.0 (further, Fintech 4.0) additional discussion is needed. Academic research is progressive, making it important to gather pertinent data and conduct further studies on how Fintech affects bank efficiency (performance). Future research may examine these aspects in the context of the Greater China region or emerging Asian economies to untangle the productivity paradox. Especially, Fintech has made significant progress in the Chinese financial market, but whether Fintech has already played a role in China's banking industry remains to be further studied. It is suggested that future research consider text mining, and weight indicators by the entropy method and grey relational analysis to measure Fintech development levels.

## Appendix

To determine causality and its direction, if any, between fintech-related initiatives variables ($ITE_t$ and $ATM_t$) and efficiency ($BEFF_t$), this study used a Vector Autoregressive Distributed lag (ADL) model specification with two lags in order to assess the direction of the relationship between IT investment, fintech-related initiatives and bank efficiency. This approach literature (Berger and Humphrey, [40]) is beneficial in that it permits examining the

intertemporal relationship between IT business services and efficiency; this study will be preoccupied with estimating the following fundamental equations below:

$$BEFF_{it} = \alpha_0 + \alpha_1 BEFF_{i,t-1} + \alpha_2 BEFF_{i,t-2} + \alpha_3 ITE_{i,t-1} + \alpha_4 ITE_{i,t-2} + \\ \alpha_5 ATM_{i,t-1} + \alpha_6 ATM_{i,t-2} + \delta_{it} + \varepsilon_{it}$$ (A.1)

$$ITE_{it} = \alpha_0 + \alpha_1 BEFF_{i,t-1} + \alpha_2 BEFF_{i,t-2} + \alpha_3 ITE_{i,t-1} + \alpha_4 ITE_{i,t-2} + \\ \alpha_5 ATM_{i,t-1} + \alpha_6 ATM_{i,t-2} + \delta_{it} + \varepsilon_{it}$$ (A.2)

$$ATM_{it} = \alpha_0 + \alpha_1 BEFF_{i,t-1} + \alpha_2 BEFF_{i,t-2} + \alpha_3 ITE_{i,t-1} + \alpha_4 ITE_{i,t-2} + \\ \alpha_5 ATM_{i,t-1} + \alpha_6 ATM_{i,t-2} + \delta_{it} + \varepsilon_{it}$$ (A.3)

where $BEFF_{it}$ indicates bank efficiency include X-efficiency (XEFF) and cost efficiency (CE), $ATM_{i,t-2}$, is lag two period number of automated teller machines, $ITE_{i,t-2}$ is lag two period information technology expense per year, $\delta_{it}$ is an individual bank specific effect and $\varepsilon_{it}$ is a disturbance term.

If one or more of the employed datasets suffers from autocorrelation, lagged regression is susceptible to over-reporting the relationship. Thus, McGraw and Barnes [41] argue that the Granger-causality approach is better than the traditionally lagged regression.

## Supporting information

**S1 Data.**
(XLSX)

## Author Contributions

**Conceptualization:** Chang-Sheng Liao.

**Data curation:** Chang-Sheng Liao.

**Formal analysis:** Chang-Sheng Liao.

**Funding acquisition:** Chang-Sheng Liao.

**Investigation:** Chang-Sheng Liao.

**Methodology:** Chang-Sheng Liao.

**Project administration:** Chang-Sheng Liao.

**Resources:** Chang-Sheng Liao.

**Software:** Chang-Sheng Liao.

**Supervision:** Chang-Sheng Liao.

**Validation:** Chang-Sheng Liao.

**Visualization:** Chang-Sheng Liao.

**Writing – original draft:** Chang-Sheng Liao.

**Writing – review & editing:** Chang-Sheng Liao.

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
