## [Decision Letter · Decision Letter 0]

12 Dec 2022

PONE-D-22-27977The Impact of Fintech on Bank Efficiency: Application of parametric and non-parametric approaches

PLOS ONE

Dear Dr. Liao,

Thank you for submitting your manuscript to PLOS ONE. In view of the referees’ feedback and my own reading of your paper, we believe your paper is some way from being publishable. In particular, it is not clear the contribution of the manuscript to the literature on this topic and there also some concerns about the robustness of the results and conclusions.

While we consider the issues identified to be major in nature, we are willing to offer you a chance to rework the paper if you feel able to address them fully and robustly.

Therefore, we invite you to submit a revised version of the manuscript that addresses the points raised during the review process.

We look forward to receiving your revised manuscript.

Kind regards,

J E. Trinidad Segovia

Section Editor

PLOS ONE

Journal Requirements:

 "NO - Include this sentence at the end of your statement: The funders had no role in study design, data collection and analysis, decision to publish, or preparation of the manuscript."

4. We note you have included a table to which you do not refer in the text of your manuscript. Please ensure that you refer to Table 9 in your text; if accepted, production will need this reference to link the reader to the Table.

Reviewers' comments:

Reviewer's Responses to Questions

**Comments to the Author**

1. Is the manuscript technically sound, and do the data support the conclusions?

Reviewer #1: Partly

Reviewer #2: Yes

Reviewer #3: No

2. Has the statistical analysis been performed appropriately and rigorously? 

Reviewer #1: No

Reviewer #2: Yes

Reviewer #3: Yes

3. Have the authors made all data underlying the findings in their manuscript fully available?

Reviewer #1: Yes

Reviewer #2: Yes

Reviewer #3: No

4. Is the manuscript presented in an intelligible fashion and written in standard English?

Reviewer #1: No

Reviewer #2: Yes

Reviewer #3: No

5. Review Comments to the Author

Reviewer #1: The Impact of Fintech on Bank Efficiency: Application of parametric and non-parametric approaches

Journal: PLOS ONE

Summary

This study investigates the determinants of bank cost efficiency in Taiwan.

After reading this paper draft, I would like to raise some major concerns for further action at your desk.

Introduction

The contribution of this study is not clearly explained.

More especially, the word “fintech” seems to be overused and misinterpreted in this manuscript unless a convincing and clear definition of fintech is presented. Therefore, this study tends to examine the impact of delivery channels such as ATMs, and bank cards on bank efficiency. Please see Le and Ngo (2020) for a more comprehensive discussion.

Literature review

The first hypothesis is not really correct. Please rewrite it properly.

Data and methodology

Data

Table 1: Please provide a brief description of the variables used. Also please provide proper descriptive statistics of variables (e.g., min, max,..).

There are missing descriptive statistics for some important regressors.

Model specification

I am not convinced that the transaction value of ATMs and credit cards is used as two additional outputs. When these delivery services increase, banks would earn more fees. Therefore, the second output (revenue fee) has already accounted for this perspective. That is why I am not surprised that the results of efficiency scores are insignificant different when these two additional outputs are included/excluded.

Please reconsider this suggestion in your revision.

In equation (8), there are a few issues here:

• EPAY or MPAY? This variable should be measured by the transaction value of the mobile payment service. Unfortunately, the dummy variable is used. However, there is a disadvantage to using this. For example, can a firm change the status of offering mobile payment service over time (e.g., static indicator, selection indicator..)? Please see Berger et al. (2005). It would be same as the case of TELEP.

• I am very confused about the definitions of regressors. Please keep them consistency.

• Non-performing loans?

Results

Regression analysis

Please provide definitions of external regressors in the note in each table.

I am not going to make further comments on this section unless these above issues are well explained. If the contribution of this study is not well articulated, the rest of this paper will look like to be a data-crunching exercise in the context of the Taiwanese market.

References

Berger, Allen N, George RG Clarke, Robert Cull, Leora Klapper and Gregory F Udell (2005), Corporate governance and bank performance: A joint analysis of the static, selection, and dynamic effects of domestic, foreign, and state ownership, Journal of Banking & Finance, 29, 2179-2221.

Le, Tu DQ and Thanh Ngo (2020), The determinants of bank profitability: A cross-country analysis, Central Bank Review, 20, 65-73.

Reviewer #2: In the methodology:

a) Does the size of the bank in terms of employees matter?

b) Could banks be using Fintech have fewer employees and higher salaries?

c) Might use of average/median salary be better than TOTAL SALARY?

d) Why not deal with banks in the same tier? (I feel like you are comparing apples and oranges)

Consider these four points in the methodology and data analysis.

In the empirical results

a) The discussion of the findings based on hypothesis 3a and 3b is missing

b) Why does DFA outperform SFA in your study? A plausible explanation would be good.

Reviewer #3: The Impact of Fintech on Bank Efficiency: Application of parametric and non-parametric approaches

Review Report

I get the chance to read this paper. It is an interesting paper which aims to determine the relationship between bank efficiency and Fintech related-business. However, the author is unable to find any significant relationship.

Introduction

I am not satisfied with the introduction session. The current version of the introduction is emphasising more about literature and the importance of Fintech firms and Taiwan's economy:

1. The important part was the contribution of the study, which I was unable to find. I would encourage the author to look into the importance of this study. Like if this study is not conducted, what would happen?

2. The? manuscript is redundant as the literature mentioned in the introduction session is also cited in the literature review session. Hence, the introduction was unable to introduce the topic.

3. The introduction part needs to include the practical implications of the topic.

The Fintech operations are beneficent for the economy and the efficiency of banks. If the shareholder is holding a zero portfolio, where did he need to invest? Should he/she invest in the banking sector after the Fintech related-businesses are commenced or before? Also, what is the impact of Fintech businesses on the overall economy? Can it help strengthen the economy and pay back the debt of Taiwan, or to what extent?

There are several studies which discuss the impact of Fintech-related services on the performance and efficiency of the banking sector (please see references below). In the last, I failed to see the gap in my study. What is new about this study? The author does not justify the novelty of the study.

Given the above reason, this paper can improve the current version of the introduction.

References:

Chen, X., You, X., & Chang, V. (2021). FinTech and commercial banks' performance in China: A leap forward or survival of the fittest?. Technological Forecasting and Social Change, 166, 120645.

Nastiti, N. D., & Kasri, R. A. (2019). The role of banking regulation in the development of Islamic banking financing in Indonesia. International Journal of Islamic and Middle Eastern Finance and Management.

Zhao, J., Li, X., Yu, C. H., Chen, S., & Lee, C. C. (2022). Riding the FinTech innovation wave: FinTech, patents and bank performance. Journal of International Money and Finance, 122, 102552.

Literature Review

The paper's literature session is a weakness in terms of theoretical background. There is no grounding theory related to this topic mentioned by the author. There are several efficiency theories pertaining to the efficiency theory and bank performance. Without proper background theory, I am unable to see the relevance of the literature review.

The hypotheses development is poorly defined, such as the first hypothesis being more appropriate for the methodology session. The author needs to realise what is testing, whether this is a Fintech-related service on the bank efficiency or a new methodology introduced by him.

I am concerned by the proxies of the Fintech related-business. The author mentioned, "Generally, fintech refers to the combination of finance and information technology, using IT-related activities to improve bank efficiencies, such as ATM, credit card, internet banking, and mobile payment.". The proxies for Fintech taken is from industry 3.0, where ATMs are considered a big development in the banking infrastructure. We are standing at Industrial revolution 4.0 and moving to industrial revolution 5.0. These proxies for Fintech-related business can not be called "Fintech" in our study now. I agree that those were significant developments around two decades back, but do these proxies have any relevance in today's world? This question is important to understand as the whole foundation of this paper depends on the proxied for the Fintech related-business. In addition, mobile payments have a positive impact on bank efficiency. Indeed, there are shreds of evidence already researched the Fintech involvement in the banking sector. The author did not emphasise what is new about this paper.

The 4th hypothesis the author mentioned "It is also a critical measure of the effect of Fintech is the bank's IT investment. To what extent have IT investments impacted firm efficiency and productivity? While studies argue that IT investments can allow firms to achieve both revenue growth and cost savings (Mithas et al. 2012), Strassman (2002) insists that IT investment cannot contribute to productivity or corporate performance as expected. Investment in IT has grown remarkably; however, the growth of productivity in individual firms and in the entire economy has been relatively slow. Beccalli (2007) found little relationship between IT investment and improved bank profitability or efficiency, indicating the existence of a profitability paradox. Gupta et al. (2018) confirm the profitability paradox in the Indian banking sector. This phenomenon raises doubts regarding IT's ability to fulfil benefits that once seemed promising (Rao and Mandal 2012). Ou et al.(2009) found that IT investment is associated with positive bank efficiency in Taiwan. However, numerous studies cannot appropriately explain the performance resulting from IT investments"

The references are from 2012 (one decade back) and more; several studies were conducted during the last decade on this topic. Considering expenditure on IT equipment is not an appropriate proxy for the Fintech firms. The firms may buy new computers rather than a cybersecurity system (Fintech). I would recommend removing this hypothesis better to understand the Fintech businesses with the efficiency of banks.

Methodology:

The authors do a good job in this session. They ascertain the efficiency of the bank by using the DFA, SAF, and DEA approaches. Then, they run the t-test and one-way ANOVA for all these three models. Afterwards, the author used these proxies for the regression models and saw the impact of Fintech services (proxied as mentioned above) on the efficiency of banks. However, the regression models (Table 9) do not control for other bank-level factors, such as leverage or ownership structure.

For the robustness test, the authors should use an alternative proxy for the Fintech firms and efficiency. Such as, there are several efficiency ratios at the bank level. I suggest reading the recent paper by Dhiaf et al. (2022). The paper examines the impact of Fintech firms on manufacturing efficiency. I recommend to the author that he can take similar proxies for the robustness test.

Reference:

Dhiaf, M. M., Khakan, N., Atayah, O. F., Marashdeh, H., & El Khoury, R. (2022). The role of FinTech for manufacturing efficiency and financial performance: in the era of industry 4.0. Journal of Decision Systems, 1-22.

Conclusion:

The conclusion session is up to a satisfactory level, but it needs the future studies part of the conclusion, like how future studies can improve the current version of this paper and can contribute to it.

6. PLOS authors have the option to publish the peer review history of their article (what does this mean?). If published, this will include your full peer review and any attached files.

Reviewer #1: No

Reviewer #2: No

Reviewer #3: **Yes: **Khakan Najaf

---

## [Author Response · Author response to Decision Letter 0]

17 Mar 2023

Dear Editor:

Thank you for your valuable suggestions.

I also appreciate the time and effort you and each reviewer have dedicated to providing insightful feedback on ways to strengthen my paper. Thus, it is with great pleasure that we resubmit our article for further consideration. I have made major changes that reflect the detailed suggestions you gave me. 

In order to make the research topic of this study easier to understand and focus it. I change the manuscript title to the new title: How Does Fintech Affect Bank Efficiency in Taiwan? 

Thank two anonymous referees for their valuable suggestions and problems. To facilitate your review of our revisions, I've included a point-by-point response to the questions and comments delivered in your letter. Please look “Response to Reviewers" file. Again, thank you for giving me the opportunity to strengthen my manuscript with your valuable comments and queries. I have worked hard to incorporate your feedback and hope these revisions persuade you to accept my submission. 

I hope to hear from you soon. 

Thanking you in anticipation,

Best regards,

Chang-Sheng Liao

School of Economics and Management, Hubei Polytechnic University, Hubei, China. 

Department of Finance and Cooperative Management, National Taipei University, Taiwan, Republic of China

---

## [Decision Letter · Decision Letter 1]

30 May 2023

PONE-D-22-27977R1How Does Fintech Affect Bank Efficiency in Taiwan?PLOS ONE

Dear Dr. Liao,

Thank you for submitting your manuscript to PLOS ONE. After careful consideration, we feel that it has merit but does not fully meet PLOS ONE’s publication criteria as it currently stands. Therefore, we invite you to submit a revised version of the manuscript that addresses the points raised during the review process.

There is no doubt that the authors have made a substantial effort to improve this manuscript and this has been appreciated by the reviewers. In fact, one of them recommends accepting the publication of it in its current state. However, the other reviewer has have identified a few areas that require some further attention to ensure the manuscript meets the high standards of PLOS ONE. These include explicitly mentioning the limitations of the study, giving more detailed recommendations for future research, stating the study's contribution to existing literature more clearly, providing more in-depth practical implications, and including a synthesis of the hypotheses testing in the conclusion.

Despite the fact that the reviewer understands these changes as major, for my part I believe that they do not require excessive effort on the part of the authors and I believe that once they have been made, the reviewer will give his approval for acceptance.

For my part, I would like to indicate that the work carried out by the reviewers has been exceptional and I believe that the manuscript has greatly benefited from it.

We look forward to receiving your revised manuscript.

Kind regards,

J E. Trinidad Segovia

Section Editor

PLOS ONE

Journal Requirements:

Reviewers' comments:

Reviewer's Responses to Questions

**Comments to the Author**

1. If the authors have adequately addressed your comments raised in a previous round of review and you feel that this manuscript is now acceptable for publication, you may indicate that here to bypass the “Comments to the Author” section, enter your conflict of interest statement in the “Confidential to Editor” section, and submit your "Accept" recommendation.

Reviewer #2: All comments have been addressed

Reviewer #3: (No Response)

2. Is the manuscript technically sound, and do the data support the conclusions?

Reviewer #2: Yes

Reviewer #3: Yes

3. Has the statistical analysis been performed appropriately and rigorously? 

Reviewer #2: Yes

Reviewer #3: Yes

4. Have the authors made all data underlying the findings in their manuscript fully available?

Reviewer #2: Yes

Reviewer #3: Yes

5. Is the manuscript presented in an intelligible fashion and written in standard English?

Reviewer #2: Yes

Reviewer #3: Yes

6. Review Comments to the Author

Reviewer #2: The authors worked on the comments raised, particularly in the methodology section of the paper. The parametric and non-parametric DEA models (DFA and SFA) are analyzed and compared.

Reviewer #3: Reviewer Report

Title: The Impact of Fintech on Bank Efficiency: Application of parametric and non-parametric approaches

The submitted paper explores a highly relevant topic in the current banking and finance environment: the impact of fintech on bank efficiency. It brings a case study of Taiwan to investigate the relationship between IT investment, fintech outputs, and bank efficiency.

General Evaluation:

The paper seems to be well-structured and provides a good overview of the topic at hand. However, several improvements could significantly enhance the impact and clarity of the paper.

Significance:

The topic is indeed significant, given the rapid growth of the fintech industry and its potential implications for traditional banking operations. Investigating the impact of this phenomenon on bank efficiency is a worthy area of study.

Contribution & Novelty:

The study appears to contribute to existing literature by looking at the impact of fintech on bank efficiency, which seems to be less explored, especially concerning the Taiwan case. It is also positive to see that the study is utilizing both parametric and non-parametric approaches, contributing to the diversity of methodological approaches in this area.

Research Gap:

The introduction identifies a gap in the literature on the influence of fintech on bank efficiency, emphasizing that past studies have provided inconclusive evidence on the link between IT investment and efficiency. However, the identified gap is not clearly stated or strongly justified. The authors should better articulate the gap by referring to more specific limitations of the prior studies.

Structure:

The structure of the paper is well-organized, with distinct sections for introduction, literature review, model specification, empirical results, and conclusions. However, the introduction part seems rather long. It should succinctly present the research problem, the gap, and the objectives of the study, leaving the details of fintech evolution and specific technologies for the literature review section.

Specific Comments:

1. The research problem is clear, but the research question or hypotheses are not explicitly stated. The authors should explicitly state the research questions or hypotheses that will guide their analysis.

2. The introduction provides a lot of background information on the fintech evolution in Taiwan. While it's important to understand the context, this section appears to be overly descriptive. Such detailed information might be better placed in the literature review or in a separate 'background' section.

3. It's unclear what the authors mean by the "IT productivity paradox" in the abstract. This should be clarified and further elaborated upon in the introduction or literature review sections.

4. The authors refer to several studies without clearly linking them to the research problem or gap. The literature should be used more effectively to provide support for the research problem and to illustrate the gap that this paper will address.

5. While the authors suggest that IT investments may lead to improved bank efficiency, they also mention that the effects are inconsistent. The authors should provide more detail on what they mean by this inconsistency and how their study will address this.

Recommendations for Authors:

To enhance the quality of this study, the authors should:

1. Make the research question or hypotheses explicit.

2. Trim down the historical overview of fintech in the introduction and focus more on presenting the research problem, research gap, and objectives of the study.

3. Better articulate the gap in the existing literature and clarify how this study will address it.

4. Provide more detail on the inconsistency in the effects of IT investments on bank efficiency.

5. More effectively use the referenced literature to support the research problem and illustrate the gap.

I hope the authors find these comments helpful in refining their paper. The study has potential to provide valuable insights into the impact of fintech on bank efficiency and I look forward to seeing the revised manuscript.

Literature and Hypotheses:

The Literature Review and Hypotheses section of the paper provides a detailed account of prior research on the impact of fintech and IT on bank efficiency. It offers an array of perspectives and covers a range of banking aspects, from ATMs to mobile banking. The authors have shown due diligence in mentioning conflicting opinions, which adds depth to the study. However, there are several areas for improvement and clarification.

1. Structure: The literature review appears somewhat disorganized. It is recommended to structure the literature review more effectively around the proposed hypotheses. This would enhance the clarity and flow of the arguments being made.

2. Explanation of Key Concepts: While the paper assumes some prior knowledge of the subject from the reader, it would be beneficial to include definitions or explanations of key concepts or terms such as DEA (Data Envelopment Analysis), POS (Point of Sale), and more.

3. Inconsistencies in Arguments: There is an inconsistency in the hypothesis section regarding ATM services. H2a suggests that ATM services have a positive impact on bank efficiency, while H2b suggests the opposite. The authors should provide additional clarification or separate these into distinct hypotheses based on different conditions or circumstances.

4. Validation of Hypotheses: More robust arguments and clear empirical evidence should be provided to validate the proposed hypotheses. This will enhance the credibility of the paper.

5. Contextual Background: The paper might benefit from adding contextual information about Taiwan's banking system and fintech sector, as the study's focus is on Taiwan.

6. Incorporation of Theory: For theoretical underpinning, the authors may consider using the Resource-Based View (RBV) or the Dynamic Capabilities View (DCV). These theories help explain how resources such as IT capabilities can contribute to firm efficiency and performance.

7. Typographical Errors: There are several minor typographical errors, e.g., and "Hong Kang" should be "Hong Kong". A thorough proofreading is suggested.

In conclusion, the paper provides a comprehensive review of the literature, even though it would benefit from a more structured layout, additional clarification, and validation of the hypotheses, and minor typographical corrections. Once these recommendations are implemented, the paper would significantly contribute to the literature on the impact of fintech on bank efficiency.

Methodology:

It is improved in this round of revision. However, the authors mentioned that we have mentioned VIF level in Table 6 but there is no VIF level in Table 6. Similarly, the table 7 is has mentioned only one regression test VIF, since there are couple of regression Models are ran then the authors should mentioned the both regression model VIF levels.

Conclusion:

The Conclusion section provides a comprehensive summary of the findings from the study, highlighting the effects of fintech on bank efficiency. It notes the role of excellent management in improving bank efficiency and emphasizes the need for policy initiatives, effective IT strategies, and other resources. It could be improved with the addition of certain key elements.

1. Limitations of the Study: The authors should explicitly mention the limitations of the study. This might include limitations related to data, methodology, or the scope of the study.

2. Recommendations for Future Research: While the authors mention the need for future research to examine these aspects in the context of the Greater China marketplace or emerging Asian economies, it would be beneficial to expand upon this point. For example, suggest specific research questions or hypotheses that future studies could explore.

3. Contribution to Existing Literature: The authors should state more explicitly how their study contributes to existing literature. While the authors note that their study addresses a major gap in the literature, they could expand on this point by discussing how their findings provide new insights or advance our understanding of the topic.

4. Practical Implications: The authors mention several policy and managerial implications. However, they could provide more detailed suggestions or recommendations for managers and policymakers. This could help make the paper more relevant and impactful.

5. Synthesis of Hypotheses Testing: The authors discuss the rejection of Hypothesis H1 but do not mention the results for the other hypotheses. It would be beneficial to include a synthesis of the hypotheses testing in the conclusion.

In summary, while the conclusion is substantial, it would greatly benefit from clear delineation of the study's limitations, contributions to the literature, and a more detailed discussion of recommendations for future research and practical implications. These improvements would enhance the quality and clarity of the paper.

7. PLOS authors have the option to publish the peer review history of their article (what does this mean?). If published, this will include your full peer review and any attached files.

Reviewer #2: No

Reviewer #3: **Yes: **khakan najaf

---

## [Author Response · Author response to Decision Letter 1]

13 Jul 2023

Dear Dr Najaf:

Thank you for your valuable suggestions.

I also appreciate the time and effort you and each reviewer have dedicated to providing insightful feedback on ways to strengthen my paper. Thus, it is with great pleasure that we resubmit our article for further consideration. I have made major changes that reflect the detailed suggestions you gave me. 

To facilitate your review of our revisions. I've included a point-by-point response to the questions and comments delivered in your letter. Please look “Response to Reviewers" file. Again, thank you for giving me the opportunity to strengthen my manuscript with your valuable comments and queries. I have worked hard to incorporate your feedback and hope these revisions persuade you to accept my submission.

I hope to hear from you soon. 

Thanking you in anticipation,

Best regards,

Chang-Sheng Liao

School of Economics and Management, Hubei Polytechnic University, Hubei, China. 

Department of Finance and Cooperative Management, National Taipei University, Taiwan, Republic of China

---

## [Decision Letter · Decision Letter 2]

24 Jul 2023

How Does Fintech Affect Bank Efficiency in Taiwan?

PONE-D-22-27977R2

Dear Dr. Liao,

We’re pleased to inform you that your manuscript has been judged scientifically suitable for publication and will be formally accepted for publication once it meets all outstanding technical requirements.

Kind regards,

J E. Trinidad Segovia

Section Editor

PLOS ONE

Additional Editor Comments (optional):

Reviewers' comments:

Reviewer's Responses to Questions

**Comments to the Author**

1. If the authors have adequately addressed your comments raised in a previous round of review and you feel that this manuscript is now acceptable for publication, you may indicate that here to bypass the “Comments to the Author” section, enter your conflict of interest statement in the “Confidential to Editor” section, and submit your "Accept" recommendation.

Reviewer #3: All comments have been addressed

2. Is the manuscript technically sound, and do the data support the conclusions?

Reviewer #3: Yes

3. Has the statistical analysis been performed appropriately and rigorously? 

Reviewer #3: Yes

4. Have the authors made all data underlying the findings in their manuscript fully available?

Reviewer #3: Yes

5. Is the manuscript presented in an intelligible fashion and written in standard English?

Reviewer #3: Yes

6. Review Comments to the Author

Reviewer #3: (No Response)

7. PLOS authors have the option to publish the peer review history of their article (what does this mean?). If published, this will include your full peer review and any attached files.

Reviewer #3: **Yes: **khakan naj

---

## [Editor Report · Acceptance letter]

28 Jul 2023

PONE-D-22-27977R2 

How Does Fintech Affect Bank Efficiency in Taiwan? 

Dear Dr. Liao:

I'm pleased to inform you that your manuscript has been deemed suitable for publication in PLOS ONE. Congratulations! Your manuscript is now with our production department. 

Kind regards, 

on behalf of

Dr. J E. Trinidad Segovia 

Section Editor

PLOS ONE